# EEG Signals Feature Extraction Based on DWT and EMD Combined with Approximate Entropy

**DOI:** 10.3390/brainsci9080201

**Published:** 2019-08-14

**Authors:** Na Ji, Liang Ma, Hui Dong, Xuejun Zhang

**Affiliations:** 1College of Electronic and Optical Engineering, Nanjing University of Posts and Telecommunications, Nanjing 210023, China; 2Nation-Local Joint Project Engineering Lab of RF Integration & Micropackage, Nanjing University of Posts and Telecommunications, Nanjing 210023, China

**Keywords:** classification recognition rate, discrete wavelet transform, empirical mode decomposition, intrinsic mode functions, approximate entropy

## Abstract

The classification recognition rate of motor imagery is a key factor to improve the performance of brain–computer interface (BCI). Thus, we propose a feature extraction method based on discrete wavelet transform (DWT), empirical mode decomposition (EMD), and approximate entropy. Firstly, the electroencephalogram (EEG) signal is decomposed into a series of narrow band signals with DWT, then the sub-band signal is decomposed with EMD to get a set of stationary time series, which are called intrinsic mode functions (IMFs). Secondly, the appropriate IMFs for signal reconstruction are selected. Thus, the approximate entropy of the reconstructed signal can be obtained as the corresponding feature vector. Finally, support vector machine (SVM) is used to perform the classification. The proposed method solves the problem of wide frequency band coverage during EMD and further improves the classification accuracy of EEG signal motion imaging.

## 1. Introduction

Brain–computer interface (BCI) is a communication control system that does not rely on the normal output channels of muscles and the nerves around the brain. BCI bypasses external nerves and muscle tissue, directly establishing information channels between brain consciousness and external devices [1], avoiding nerve conduction and muscle movement, providing a new means of rehabilitation for patients with nerve damage or muscle damage so that they do not have to rely on others to complete their own exercises [2]. With the development of BCI technology, paralyzed patients can use electronic devices, such as computers, neural prosthesis, and robotic arms and can also achieve other functions such as motion recovery, communication, environmental control, and even entertainment [3].

At present, BCI mainly includes five steps: Signal acquisition, preprocessing, feature extraction, feature classification, and interface device control. In the above steps, the feature signals extracted by the feature can identify the discrimination information of different imaginary motion electroencephalogram (EEG) signals. It has a great influence on the subsequent classification and recognition, so feature extraction has received extensive attention in the BCI research community.

The key to effective feature extraction methods is to improve recognition accuracy [4]. Traditional time-frequency methods include short-time Fourier transform (STFT), wavelet transform (WT), etc. However, the essence of these methods is based on Fourier transform, and this method cannot achieve good time-frequency resolution at the same time according to the Heisenberg uncertainty principle [5]. In recent years, the Hilbert–Huang Transform (HHT) has become more and more popular. As another time-frequency analysis method, it is also very suitable for analyzing nonlinear and non-stationary signals. This involves decomposing the original signal into a series of intrinsic mode functions (IMFs) through empirical mode decomposition (EMD) [6] and then performing HHT on each intrinsic mode function. Finally, the corresponding energy spectrum and marginal spectrum are calculated to classify the features. HHT does not involve the Heisenberg uncertainty principle and can be obtained in the time domain and frequency domain. It has been widely used in many signal processing fields, such as radar detection, seismic signals, and biomedical signals [7].

The physiological system has nonlinear characteristics, these characteristics may help to reveal signals’ characteristics and mechanisms. At present, most studies estimate the signal by estimating some nonlinear dynamic parameters of the signal (such as correlation dimension, Coriolis entropy, Lyapunov exponent), and some studies have used non-stationary linear discriminant analysis based on the Kalman smoother algorithm to estimate non-stationary parameters [8], which need more data points. Thus, these parameters are hard to apply in practice. The approximate entropy proposed by Pincus in 1991 [9] solved this kind of problem well. Approximate entropy requires only a short data point to estimate the random signal and determine the characteristics of the signal. It has good anti-noise and anti-interference ability. For EEG signals, they have both large random signal components and some certain frequency components, so the approximate entropy is very effective for the study of EEG signals [10]. Existing papers have compared the method of calculating the autoregressive coefficient and the approximate entropy and combining the features to other methods [11]. However, the classification accuracy of the former is not ideal. Wavelet transform and empirical mode decomposition before calculating approximate entropy can obtain more concentrated frequency signals so as to obtain more suitable frequency band signals.

In this paper, we propose an algorithm based on discrete wavelet transform (DWT) and EMD combined with approximate entropy. We did not use other multivariate signal techniques to decompose the input EEG signals, such as principal or independent component analysis, as it can cause the loss of some parts of EEG, and artificial discrimination of pseudo-components is time-consuming and unreliable. By calculating the approximate entropy of each frequency band of the original EEG signal instead of the approximate entropy of the single EEG signal, decomposing the EEG signal by EMD finds that the EMD decomposition has defects. For example, the IMFs generated in the low frequency region are not ideal, and the frequency band of the intrinsic modal function obtained for the first time is too wide [12]. To solve these problems, we decided to decompose the EEG signal by WT to obtain a series of narrow-band signals and then use EMD decomposition to obtain more concentrated signals for more suitable frequency band signals. Then, the approximate entropy was calculated, which provides an alternative way to discriminate the motion imagination so as to improve the classification and identification. The flow chart of the algorithm is shown in Figure 1. The main steps of the method are: (1) using WT to decompose the EEG signal into a narrowband signal; (2) using EMD to decompose the appropriate sub-band signal into IMFs; (3) using FFT to obtain the spectrum of the IMFs, selecting the IMFs of the spectrum in the μ and β rhythms to make up the signal matrix; (4) calculating the approximate entropy of the signal matrix as the feature vector; (5) using the support vector machine (SVM) classifier to classify the features.

The second part of the article describes the experimental dataset; the third part briefly introduces the algorithms of DWT, EMD, approximate entropy, and SVM; the fourth part introduces the results of data analysis; the fifth part summarizes the article.

## 2. Materials and Methods

### 2.1. Experimental Data

The experimental data came from BCI Competition 2008 datasets 2b data [13], including EEG data of nine subjects, all of whom were right-handed with normal or corrected normal vision. The experimental process is shown in Figure 2. 

Each experiment lasted 8 to 9 seconds, the subjects were relaxed in the first 2 seconds without any imaginative activity; at the 2nd second, a buzzing sound prompted the subjects that the experiment was about to begin, while the cross cursor was displayed on the screen for 1 second; at the 3rd second, visual cues (an arrow, randomly left or right) appeared on the screen for 1.25 seconds; at the 4th second, the subjects performed the motion imagination of the corresponding hands according to the direction of the prompt arrow. The imagination movement lasted for 3 seconds, and at the 7th second, the subjects stopped imagination and took a rest. The position of the experimental acquisition electrode is shown in Figure 3.

The electroencephalogram data of C3, Cz, and C4 electrodes were collected with Ag/AgCl electrodes. Cz was the reference electrode, and the position of C3 and C4 electrodes contained the most abundant information about imaginary hand movement. In this paper, the data of C3 and C4 channels are used for analysis. The sampling frequency is 250 Hz. The sampling data are filtered by 0.5–100 Hz bandpass filter and 50 Hz notch filter.

### 2.2. DWT 

The basic idea of WT is to use a cluster of functions to represent or approximate signals or functions. This function cluster is called the wavelet function system. It is formed by the translation and stretching of the basic wavelet. The transform coefficients can be approximated to the original signal. Wavelets can characterize the local characteristics of signals in both time and frequency domains. When using a smaller scale, the observation range in time domain is smaller, but in frequency domain it is equivalent to using high frequency for high resolution analysis, that is, using high frequency wavelet for detailed observation. When a large scale is used, the observation range in time domain is large, but in frequency domain it is equivalent to using low-frequency wavelet for overview observation [14]. The continuous wavelet transform of signal *x(t)* is defined as:(1)$${WT}_{x}(a,\tau )=\frac{1}{\sqrt{a}}\int_{-\infty }^{\infty }x(t)\psi (\frac{t-\tau }{a})dt$$
where *a* represents scale displacement, *τ* representing time displacement, and *ψ*(*i*) is a wavelet basis function, including Haar, db series, Coiflet, and so on.

EEG signals are discrete signals, so DWTs are needed for discrete wavelets. Compared with the continuous WT, the DWT is to limit the *a* and *τ* of the wavelet basis function ψ(a,τ) to discrete points, that is, the discretization of scale and displacement, and the discrete wavelet basis function is $\psi_{j,k}(t)=2^{-\frac{i}{2}}\psi (2^{-j}t-k)$, where j∈Z,k∈Z, the DWT is
(2)$$WT_{x}(j,k)=\int x(t)\psi_{j,k}^{*}(t)dt$$

According to the sampling theorem, the highest frequency of the signal is *f*_s_/2. Using the Mallat algorithm, if the signal is decomposed by L-order, the whole frequency band of the signal is decomposed into *L*+1 sub-band, that is, [0, *f*_s_/2*^L^*^+1^] [*f*_s_/2*^L^*^+1^, *f*_s_/2*^L^*] … [*f*_s_/2^2^, *f*_s_/2]. Figure 4 illustrates the process of wavelet decomposition by taking three-layer wavelet decomposition as an example.

### 2.3. EMD

EMD can decompose the signal into different IMFs. The basis function of EMD is determined by the signal itself. It has good adaptability and can accurately represent the signal [15]. IMFs must satisfy two conditions: One is that the maximum value of the signal amplitude is equal to the number of zero crossings in the entire signal time domain, or the maximum difference is 1, and the average value of the envelope formed by the maximum and minimum amplitude of the signal is zero.

The specific steps of EMD are as follows:

Step 1: Identify the maximum and minimum values of the signal *x*(*t*).

Step 2: For the maximum value, use the cubic spline interpolation function to fit the upper envelope *e_max_*(*t*); for the minimum value, use the cubic spline interpolation function to fit the lower envelope *e_min_*(*t*).

Step 3: Calculate the mean *m*(*t*) between the two envelopes as
(3)$$m(t)=\frac{e_{\max}(t)+e_{\min}(t)}{2}$$

Step 4: Subtract the mean from the signal to obtain the modal function *c*(*t*) as
(4)c(t)=x(t)−m(t)

Step 5: Verify whether *c*(*t*) satisfies the condition of the intrinsic mode function. If *c*(*t*) satisfies, it is an IMF component, and the original signal becomes *x_n_*_+1_(*t*):(5)$$x_{n+1}(t)=x_{n}(t)-c(t)$$

If not, go back to step 3.

Step 6: When the result signal is less than two extremum points, the residual component *r*(*t*) is reserved and the decomposition ends. The original signal is decomposed into n IMFs and the residual component is *r*(*t*):(6)$$x(t)=\sum_{i=1}^{n}c_{i}(t)+r(t)$$

### 2.4. Approximate Entropy

Approximate Entropy is the use of non-negative numbers to represent the predictability of data before and after to quantitatively describe the repeatability of time series. The more complex the approximate entropy of time series is, the more unstable the signal is, the more abundant the frequency component is, and the more complex the system is. However, the lower the approximate entropy is, the more periodic and stable the signal tends to be and the narrower the signal spectrum is [16].

Next, the definition of approximate entropy is explained by combining the algorithm steps. Let the original data be *x*(1), *x*(2), …, *x*(N), a total of *N* points.

Step 1: Form a set of *m*-dimensional vectors in sequence of numbers:(7)X(i)=[x(i),x(i+1),⋯,x(i+m−1)]
where *i* = 1~*N* − *m* + 1;

Step 2: Define the distance between *X*(*i*) and *X*(*j*). d[*X*(*i*),*X*(*j*)] is the one with the largest difference between the two elements, namely:(8)$$d[X(i),X(j)]=\underset{0\leq k\leq m-1}{\max}[|x(i+1)-x(j+k)|]$$

Step 3: Given a threshold *r*, d[*X*(*i*),*X*(*j*)] of each *i*-value statistic is less than the number of *r* and the ratio of this number to the total number of distances *N* − *m* + 1 is recorded as $C_{i}^{m}\left(r\right)$, namely:(9)$$C_{i}^{m}\left(r\right)=\mathrm{Num}\{d[X(i),X(j)]<r\}/(N-m+1)$$
where *i* = 1~*N* − *m* + 1;

Step 4: First take the logarithm of $C_{i}^{m}\left(r\right)$ and then find its average value for all *i*, denoted as $\phi^{m}(r)$, namely:(10)$$\phi^{m}(r)=\frac{1}{N-m+1}\sum_{i=1}^{N-m+1}\ln C_{i}^{m}(r)$$

Step 5: Add 1 dimension to *m* + 1, and repeat steps 1 to 4 to obtain $C_{i}^{m+1}\left(r\right)$ and $\phi^{m+1}(r)$;

Step 6: In theory, the approximate entropy of the sequence is
(11)$$ApEn(m,r)=\underset{N\to \infty }{\lim}[\phi^{m}(r)-\phi^{m+1}(r)]$$

In general, this limit exists as probability 1, but in reality *N* cannot be. When *N* is a finite value, according to the above steps, when the sequence length is *N*, an estimated value of the approximate entropy value *ApEn* is obtained, which is recorded as
(12)$$ApEn(m,r)=\phi^{m}(r)-\phi^{m+1}(r)$$

*ApEn* is obviously related to the value of *m*, *r*. According to experience, usually take *m* = 2, *r* = 0.1 ~ 0.25SD(*u*), (SD represents the standard deviation of the sequence {*u*(*i*)}), at this time, approximate entropy has more reasonable statistical characteristics [17]. The approximate entropy algorithm results reflect the complexity of the analyzed signal, regardless of the amplitude of the signal [18].

The traditional approximate entropy algorithm has redundant steps in the calculation process, which have high computational cost and low computational efficiency. Therefore, Pincus gives a fast algorithm based on the approximate entropy according to practice, which makes the calculation speed several times higher than the original algorithm. The article uses the fast algorithm of the literature [17].

### 2.5. SVM

SVM [19] achieves better separation of feature vectors by mapping low-dimensional signals to high-dimensional feature space, using hyperplane to classify data, such as two-dimensional plane data can be divided by a straight line, three-dimensional data need to be divided by two-dimensional plane, that is to say, *N*-order high-dimensional data space is divided by *N*-1-order hyperplane. When choosing the optimal classification surface, we need to be able to separate data correctly and achieve the maximum classification interval, that is, the distance from the nearest data point to the hyperplane should be as large as possible.

The classification performance of SVM is mainly affected by kernel parameters and error penalty factors, which affect the projection distribution in data space, and penalty factors determine the fault tolerance of SVM. Therefore, the selection of nuclear parameters and penalty factors has a great impact on the recognition rate of EEG signals.

In this paper, we use grid optimization and tri-fold cross-validation to optimize the selection of kernel parameters and penalty factors. We use some data as training set, change the values of kernel function and penalty factor in a certain range, and the cross-validation method is used to classify. We select the kernel parameters and penalty factors with the highest classification accuracy as the best parameters. 

## 3. Results

### 3.1. Signal Pre-Process Results

When the subject performs limb motion imaging, the sensorimotor cortex energy will change. It will generate mu rhythm and beta rhythm. Mu and beta rhythms are enhanced with event-related synchronization (ERS) during this process whereas these two rhythms are declined, which is labeled as event-related desynchronization (ERD) [20].

From the above phenomena, it can be seen that the signals with the greatest correlation with different motion imaging tasks are in the frequency range of 8–32 Hz. Most signals below 5 Hz are artifacts, so first the preprocessing is to further remove artifacts present in the signal by an 8–32 Hz bandpass filter. The c3 and c4 channel original brainwaves signal obtained by the two motion imaging types from the subject B01 were selected to obtain the preprocessing results and the Fourier transform spectrum. As shown in Figure 5a,b, the filtered signal mainly included μ rhythm and the β rhythm range.

### 3.2. DWT Results

From the prior knowledge of EEG signals, the signal frequency generated by the brain during motion imaging is mainly concentrated below 30 Hz, of which 5 Hz or less are mostly artifact signals. Therefore, the useful frequency band in EEG signal data is between 8 and 30 Hz. This paper uses db4 wavelet to perform 4th order wavelet decomposition to obtain low frequency sub-band cA_L_ and high frequency sub-band cD_L_, cD_L-1_, … cD_1_.

As shown in Table 1, sub-bands D4 and D3 are in the EEG signal band, and these two sub-band signals are selected for signal reconstruction.

As shown in Figure 6a,b, the sub-band signal frequency band is between 8 and 30 Hz, and the frequency has obvious protruding peaks, indicating that the frequency is mostly concentrated in a very narrow frequency band, which is in preparation for subsequent EMD decomposition.

### 3.3. EMD Results

WT is performed prior to EMD to split the signal into a set of narrowband signals, and the appropriate sub-band signal is selected to decompose it into an intrinsic mode function with a more concentrated frequency.

The most prominent rhythms that characterize motion imaging are μ and β rhythm. The frequency range of the two is 7~13 Hz and 13~30 Hz. The sub-band signal after the wavelet decomposition is decomposed by EMD. As shown in Figure 7 and Figure 8, the first two orders of the IMF contain the μ and β rhythm bands, and the remaining IMFs are low frequency signals which are mostly artifact signals and noise signals. Therefore, this paper only selects the first two orders of IMF to reconstruct the EEG signal and discards the rest of the IMFs. The use of EMD not only extracts the μ and β rhythms but also effectively removes the unrelated signals and increases the signal-to-noise ratio.

### 3.4. Approximate Entropy Results

Before using the EMD, we change the data during the original data extraction—extracting the C4 channel data of the left-hand experimental data and extracting the C3 channel data of the right-hand experiment. We perform wavelet decomposition on the data of the C3 and C4 channels to obtain a series of narrow-band signals. Then EMD is used to decompose the appropriate sub-band signals to obtain multiple eigenmode functions with the first two orders of μ and β rhythms taken as new input signals. This is equivalent to a total of eight channels of data, forming an 8×2000 matrix ***X_i_*** (*i* = L means imagining the movement of left hand, *i* = R means imagining the movement of right hand). The original EEG signal is decomposed by wavelet and the two sub-band signals D3 and D4 are selected, and the first two orders of IMFs are selected after EMD, which makes up a total of eight channels. 2000 is the number of sampling points of one test. When calculating the approximate entropy, we take *r* = 0.25SD(*u*), the time sliding window is used, the window length is 500 data points, and each time one data point is moved, a total of 1500 approximate entropies are obtained. The approximate entropy values of eight IMFs in the same sliding window are taken as a set of features vectors for classification.

## 4. Discussion

In this paper, we propose an EEG feature extraction method based on DWT and EMD combined with approximate entropy. After the C3 and C4 channel data of each test are decomposed by wavelet, the sub-band signal is selected, and the IMF is selected by EMD. The first two orders of IMF components are used as observation vectors to further verify the validity of the observation signal. Figure 9a,b brain topographic map of the first two orders of IMF components of the EEG signal in a single trial when subject 01 has left- and right-hand motion imaginary motion. It can be seen from the figure that when two types of imaginary movements are performed, the C3 and C4 electrodes have large energy differences. At the same time, when the subject 01 imagines the left-hand movement, it is obvious that the C4 electrode is more active than the C3 electrode. When he imagines the right-hand movement, the result is just the opposite. This coincides with the relevant synchronous desynchronization feature in the motion imagination. It is thus shown that this feature can effectively represent the spatial location information of the “source” and can be used as a classification feature. By calculating the approximate entropy, we can provide an optional way for judging the motion imagination to achieve the purpose of improving the classification recognition.

In the method of this paper, after the original EEG signal is wavelet transformed, the appropriate sub-band is selected for EMD decomposition so as to obtain multiple intrinsic mode functions IMF and then calculate the approximation of data composed of eight IMFs sampling points. Entropy, and as a feature, is classified using the SVM classifier. Figure 10 shows the classification accuracy and overall average classification accuracy of the nine trials.

Secondly, as shown in Table 2, we compare the method of this paper with other methods and list the four groups with better scores in the third BCI competition. The first group proposed a multi-feature combination method to classify these data. The three eigenvectors based on motion-related potential and event-related desynchronization are extracted by the common spatial subspace decomposition algorithm and the waveform mean and then reduced to one dimension by Fisher discriminant analysis, and they are connected into a three-dimensional feature vector. Classification is performed using a linear support vector machine. The second group used SVMs and L1 regularized logistic regression as classification algorithms. The features they chose were: Autoregressive coefficients (order 2, re-estimated every second), level 2 Haar wavelet decomposition, and spectral power estimates from 0 to 45 Hz estimated every second. Features that the third group used were mean values of selected channels in some periods of time and mean power of selected frequency intervals for selected channels. These features were constructed by hand with help of visualization of differences for the given classes. Classification is performed using logistic regression classifiers. The forth group proposed a hybrid method consisting of multivariate empirical mode decomposition (MEMD) and short time Fourier transform (STFT) to identify left- and right-hand imaginary movements from EEG signals, and their findings suggest that *k*-nearest neighbors (KNN) emerges as the best classification model. In the case of using the same dataset, the classification accuracy of the proposed method is higher than the other four groups. Under the condition of ensuring certain classification accuracy, this paper only uses the signals of C3 and C4 channels for analysis, which greatly reduces the number of channels.

In addition, as shown in Table 3, by calculating the time complexity, we found that the average classification response time of nine subjects was between 0.21 and 0.25 s. The method proposed in this paper provides feasibility for the online acquisition of signals from portable BCI systems.

Finally, in order to better verify the effectiveness of the proposed method, we also apply this method to BCI competition IV dataset 2A [21], as shown in Table 4. We compare the method proposed in this article with that of other published results. In the case of the same dataset, the average classification of the method proposed in this article is higher than that of other methods.

## 5. Conclusions

In order to solve the problem of wide frequency band coverage during EMD and further improve the classification accuracy of EEG signal motion imaging, we propose an EEG feature extraction method based on DWT and EMD combined with approximate entropy. The experimental results show that the proposed method is feasible and effective, but there is an inconsistency in the statistics of the approximate entropy. Therefore, there is still much room for improvement in accuracy. 

## Figures and Tables

**Figure 1 brainsci-09-00201-f001:**
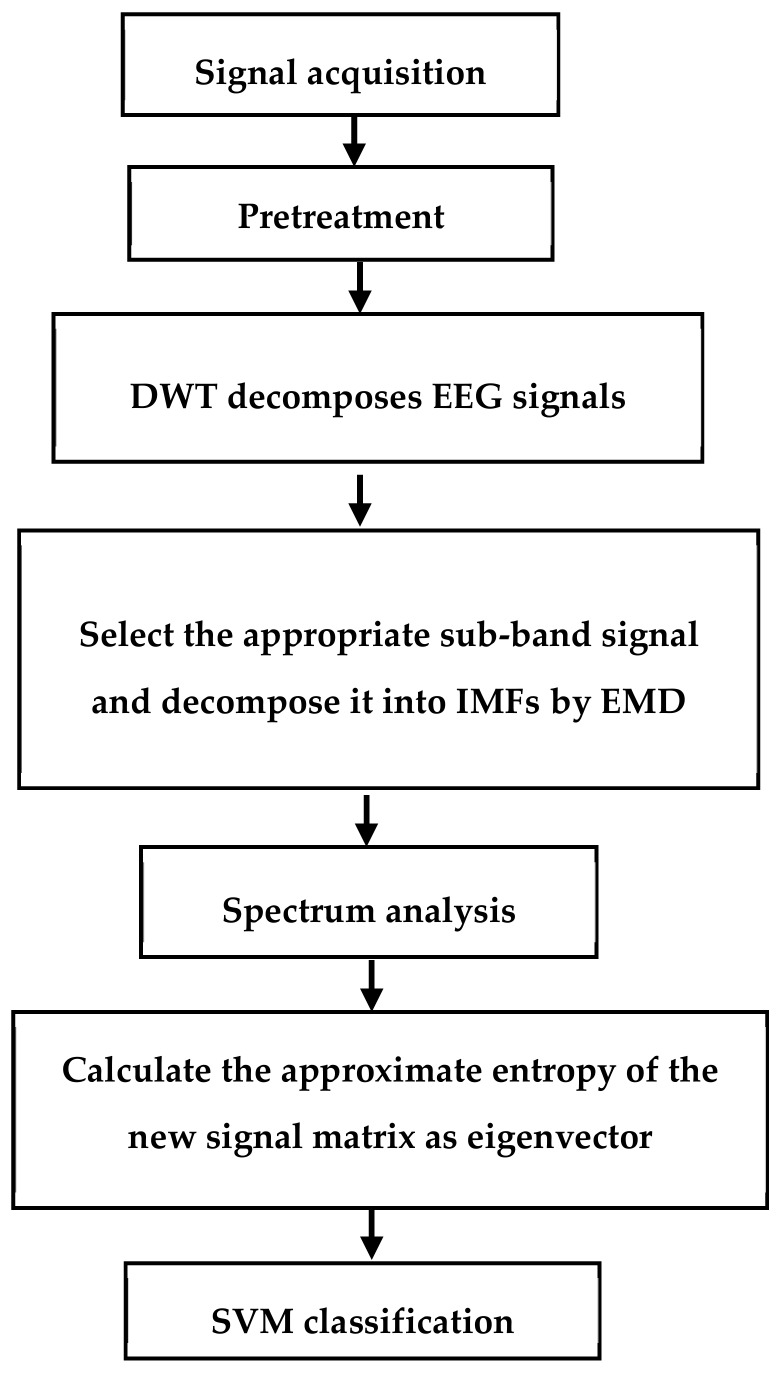
Algorithm flowchart.

**Figure 2 brainsci-09-00201-f002:**
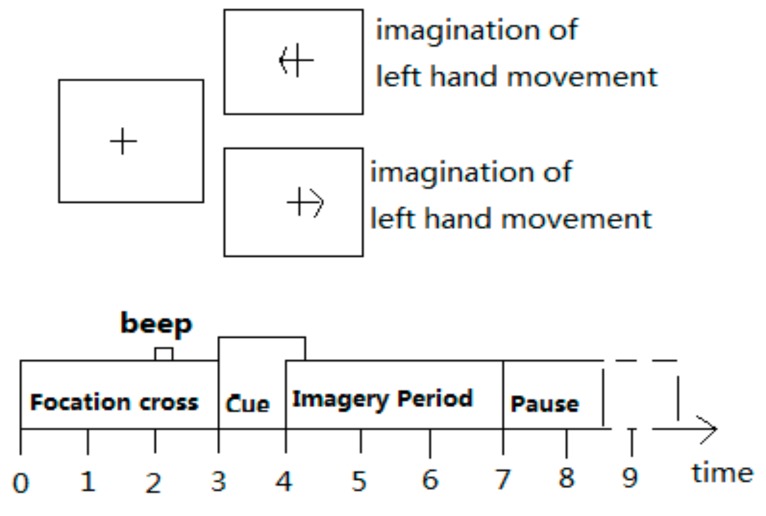
Experimental process.

**Figure 3 brainsci-09-00201-f003:**
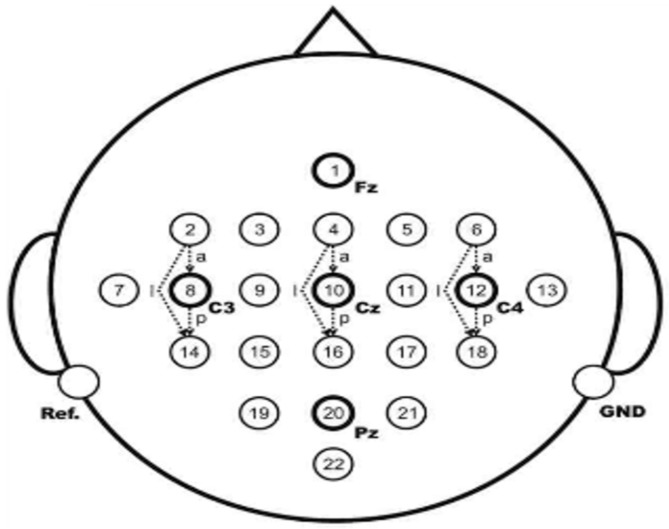
Sample electrode position.

**Figure 4 brainsci-09-00201-f004:**
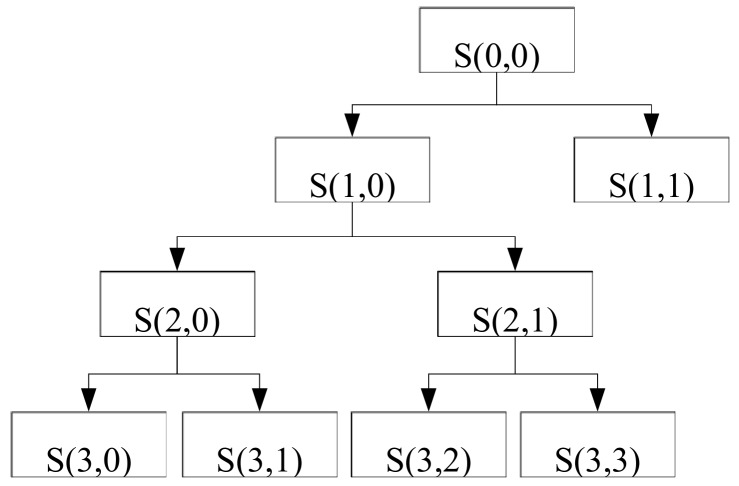
Wavelet analysis diagram.

**Figure 5 brainsci-09-00201-f005:**
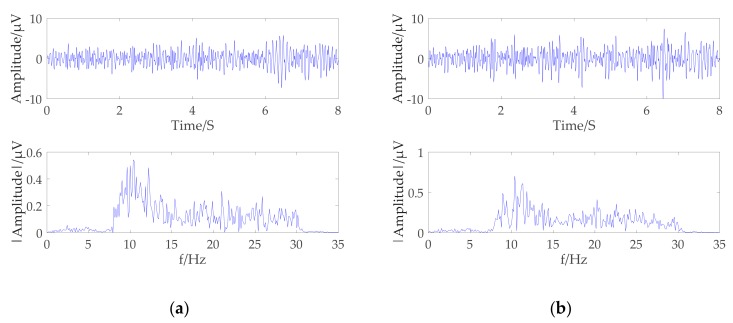
Electroencephalogram (EEG) signal and spectrum of left-hand motor imagination: (**a**) Description of C3 channel EEG signal and spectrum; (**b**) Description of C4 channel EEG signal and spectrum.

**Figure 6 brainsci-09-00201-f006:**
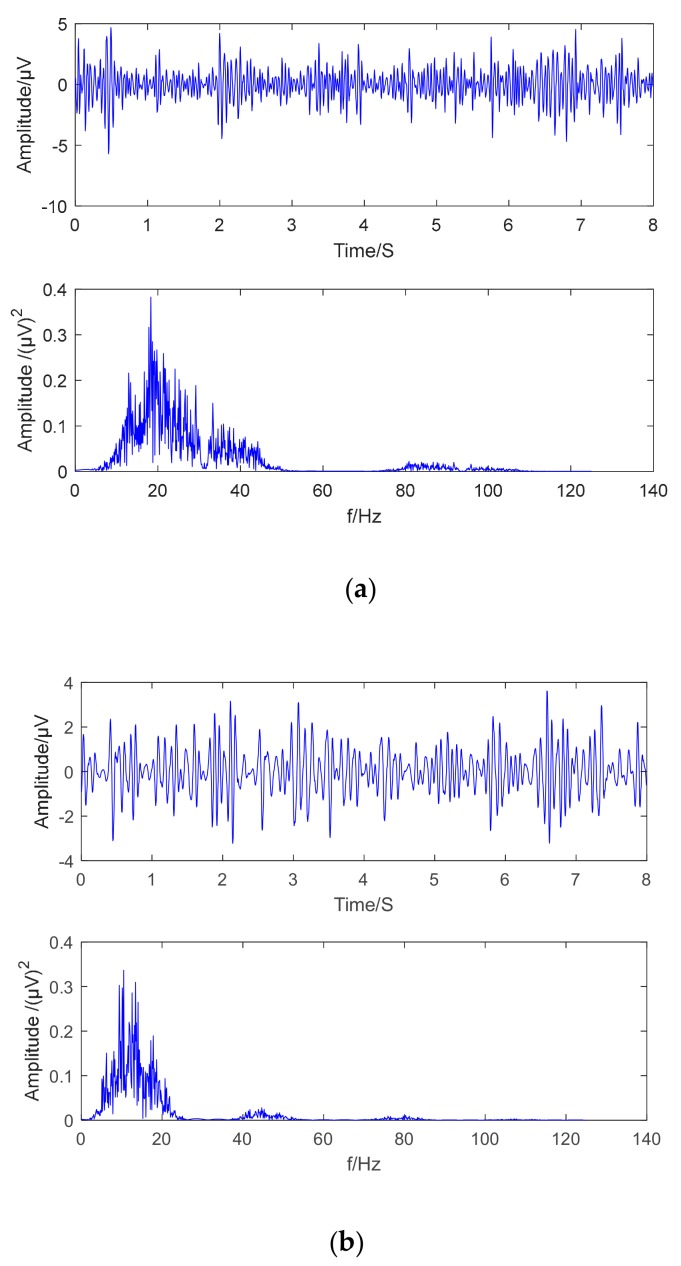
Wavelet decomposition sub-band signal (upper waveform, lower spectrum): (**a**) Wavelet decomposition sub-band signal D3; (**b**) Wavelet decomposition sub-band signal D4.

**Figure 7 brainsci-09-00201-f007:**
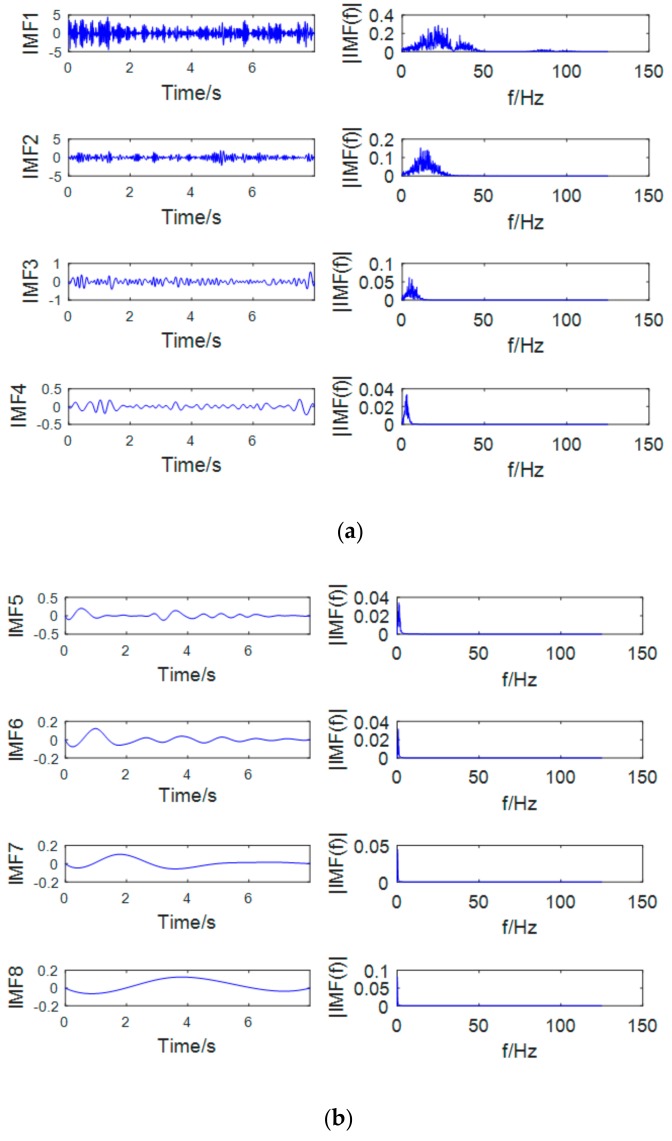
Sub-band D3 empirical mode decomposition (EMD) graph (left—waveform, right—spectrum): (**a**) IMFs and corresponding spectrum of orders of 1-4 by EMD method. (**b**) IMFs and corresponding spectrum of orders of 5-8 by EMD method.

**Figure 8 brainsci-09-00201-f008:**
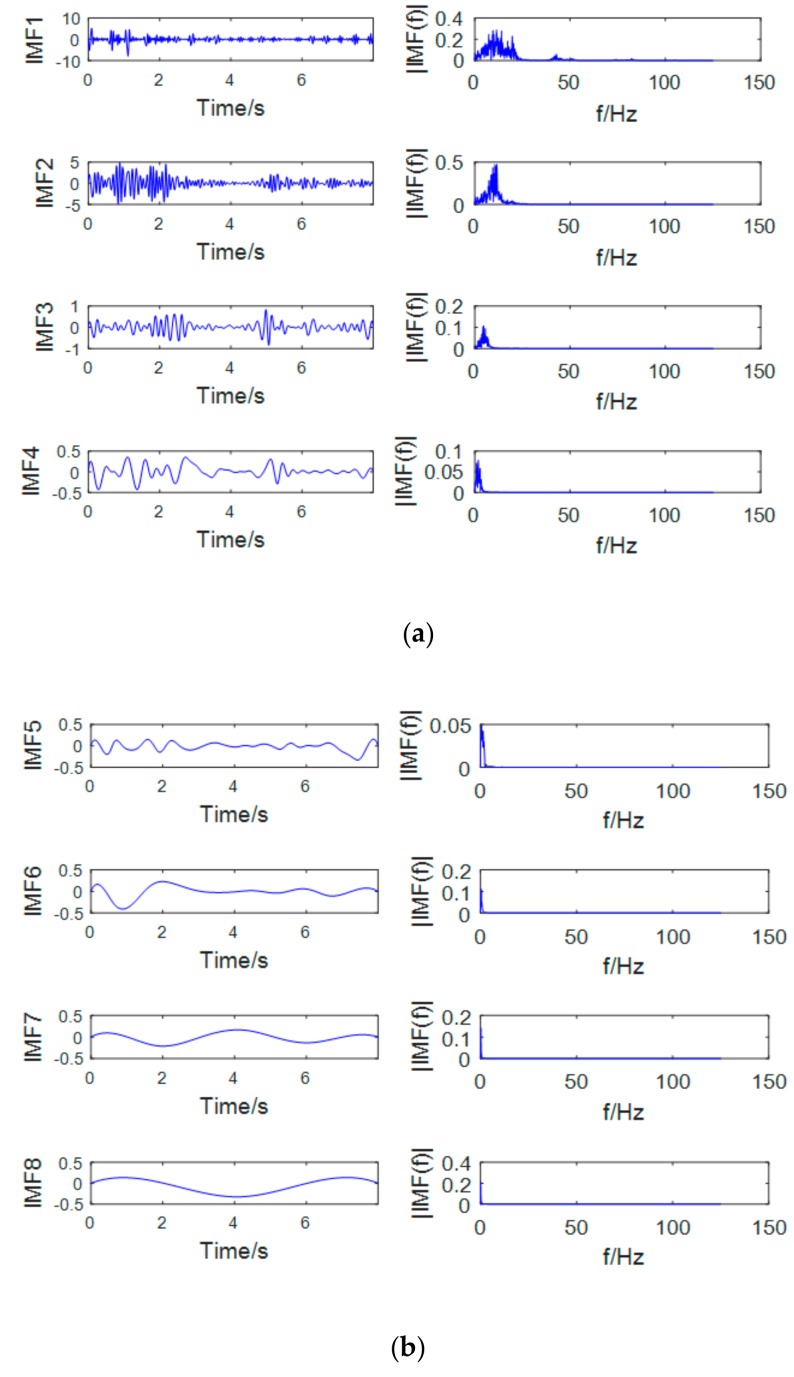
Sub-band D4 EMD graph (left—waveform; right—spectrum): (**a**) IMFs and corresponding spectrum of orders of 1-4 by EMD method; (**b**) IMFs and corresponding spectrum of orders of 5-8 by EMD method.

**Figure 9 brainsci-09-00201-f009:**
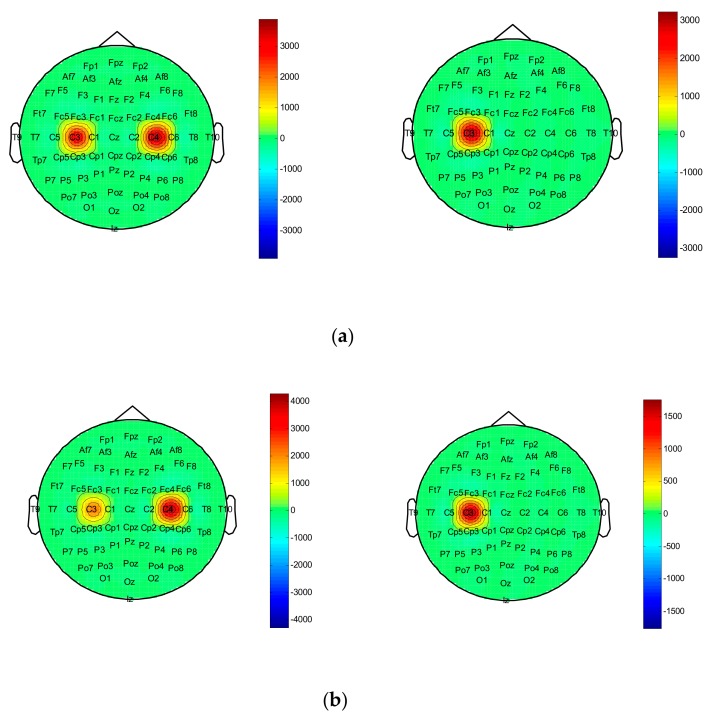
Intrinsic mode function (IMF) energy brain topographic map: (**a**) Brain topographic map of the first-order IMF energy (left—left hand, right—right hand); (**b**) Brain topographic map of the second-order IMF energy (left—left hand, right—right hand).

**Figure 10 brainsci-09-00201-f010:**
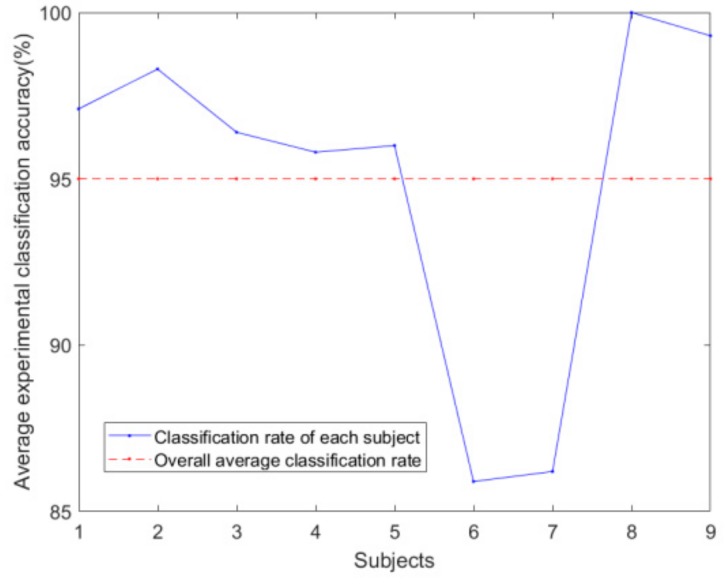
Classification accuracy comparison.

**Table 1 brainsci-09-00201-t001:** Distribution of each sub-band frequency band.

Sub-band	A4	D4	D3	D2	D1
Frequency/(Hz)	0–7.8	7.8–15.6	15.6–31.2	31.2–62.4	62.4–124.8

**Table 2 brainsci-09-00201-t002:** Comparison of brain–computer interface (BCI) competition results and the results of this paper.

	Method	Channel	Classifier	Result
Qingguo Wei [22]	CSSD + waveform mean + FDA	64	SVM	91%
Hammon [23]	AR + spectral power + wavelet coefficients	33	Regularized Logistic Regression	87%
M Sapinski [24]	Offset + spectral power	Not given	Logistic Regression	86%
SK Bashar [25]	MEMD + STFT	Not given	KNN	90.71%
Author	Proposed method	2	SVM	95.1%

**Table 3 brainsci-09-00201-t003:** Average modeling time for nine subjects.

Subjects	B01	B02	B03	B04	B05	B06	B07	B08	B09
Modeling time(s)	0.23	0.22	0.23	0.21	0.23	0.25	0.23	0.25	0.22

**Table 4 brainsci-09-00201-t004:** Comparison of classification results for the proposed method with other published results applied on BCI competition IV dataset 2A.

Studies	Wang et al. [26]	Gaur et al. [27]	She et al. [28]	Author
Method	AX-LSTM	SS-MEMDBF	FDDL-ELM	Proposed method
Result	79.6%	82.28%	80.68%	85.71%

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
