# Peer review of "EEG Signals Feature Extraction Based on DWT and EMD Combined with Approximate Entropy"

_brainsci, 2019, doi:10.3390/brainsci9080201_

Round 1

Reviewer 1 Report

The manuscript has significantly improved. A few more points needs to be addessed:

- Section 3.1: Please rephrase the sentence: "Mu and beta rhythms are enhanced during this process is event-related synchronization (ERS) whereas these two rhythms are declined is labeled as event related desynchronization (ERD) [18] ". Some linking conjunctions are missing
- Settings for approximate entropy are not reported for the threshold r. To my understanding, , i.e. length of the sequences m is equal to 500 points, that shoud be indicated more explicitely in Section 3.4 (m=500)
- Points 1, 5, 7, still need to be addressed with some quantitatification in the manuscript
- What is the time complexity of the other methods described in the paper?

Author Response

Response to Reviewer 1 Comments

Point 1: Section 3.1: Please rephrase the sentence: "Mu and beta rhythms are enhanced during this process is event-related synchronization (ERS) whereas these two rhythms are declined is labeled as event related desynchronization (ERD) [18] ". Some linking conjunctions are missing.

Response 1: We rephrased the sentence in the revised manuscript, as follows, "Mu and beta rhythms are enhanced with event-related synchronization (ERS) during this process whereas these two rhythms are declined which is labeled as event-related desynchronization (ERD)."

Point 2: Settings for approximate entropy are not reported for the threshold r. To my understanding, i.e. length of the sequences m is equal to 500 points, that shoud be indicated more explicitly in Section 3.4 (m=500)

Response 2: We take r=0.25SD(u) (SD represents the standard deviation of the sequence {u(i)}), which have been reported in the revised manuscript.

Point 3: Points 1, 5, 7, still need to be addressed with some quantification in the manuscript

Response 3: Points 1, 5, 7 were addressed in the revised manuscript.

Point 4: What is the time complexity of the other methods described in the paper?

Response 4: The time complexity of the other methods described in this paper was not reported in the corresponding literature, but the method proposed in this paper provides feasibility for the online acquisition of signals from portable BCI systems by calculating the time complexity.

Reviewer 2 Report

The paper discusses an important subject. The paper is well written and organized. My comments are as follows:

1- The introduction needs to be rewritten. It is short.

2- The main contribution of the paper is not well described in the abstract. The authors need to rewrite the abstract by emphasizing the main contribution more clearly.

3- I highly encourage the authors to include some of the recently published papers in their introduction. In particular:

- "Classification of EEG signals based on autoregressive model and wavelet packet decomposition." Neural Processing Letters 45, no. 2 (2017): 365-378.

- "Bayesian Control of Large MDPs with Unknown Dynamics in Data-Poor Environments”, Advances in Neural Information Processing Systems, pp. 8146-8156, 2018

- “MFBO-SSM: Multi-Fidelity Bayesian Optimization for Fast Inference in State-Space Models”, AAAI, 2019.

- “Nonstationary Linear Discriminant Analysis”, Proceedings of the 51st Asilomar Conference on Signals, Systems, and Computers, At Pacific Grove, CA, 2018.

4- Adding a diagram can help the readability of the paper.

5- The format of some the references is not in standard form. These need to be fixed.

Author Response

Response to Reviewer 2 Comments

Point 1: The introduction needs to be rewritten. It is short.

Response 1: We rewrote the introduction in the revised manuscript.

Point 2: The main contribution of the paper is not well described in the abstract. The authors need to rewrite the abstract by emphasizing the main contribution more clearly.

Response 2: The main contribution was emphasized in the abstract in the revised manuscript.

Point 3: I highly encourage the authors to include some of the recently published papers in their introduction. In particular:

- "Classification of EEG signals based on autoregressive model and wavelet packet decomposition." Neural Processing Letters 45, no. 2 (2017): 365-378.

- "Bayesian Control of Large MDPs with Unknown Dynamics in Data-Poor Environments”, Advances in Neural Information Processing Systems, pp. 8146-8156, 2018

- “MFBO-SSM: Multi-Fidelity Bayesian Optimization for Fast Inference in State-Space Models”, AAAI, 2019.

- “Nonstationary Linear Discriminant Analysis”, Proceedings of the 51st Asilomar Conference on Signals, Systems, and Computers, At Pacific Grove, CA, 2018.

Response 3: After listening to your suggestions, we included some of the recently published papers that you suggested in our introduction.

Point 4: Adding a diagram can help the readability of the paper.

Response 4: We added the flow chart of the algorithm in Figure 1 in the revised manuscript.

Point 5: The format of some the references is not in standard form. These need to be fixed.

Response 5: The format of some the references was fixed in the revised manuscript.

Round 2

Reviewer 1 Report

The current version of the paper is fine to me and my last comments have been addressed.

Reviewer 2 Report

The paper is well-revised.

This manuscript is a resubmission of an earlier submission. The following is a list of the peer review reports and author responses from that submission.

Round 1

Reviewer 1 Report

A classification algorithm applied to motor imagery is proposed in this article. The method results from the combination between wavelet analysis, empirical mode decomposition and entropy analysis. Although the topic is interesting, some aspects of the paper are not sufficiently detailed, especially in the Results’ section. Also, the methodology does not seem novel and the improvement over the state of the art seems very limited (if not significant at all). Please find other comments below:

-        Have the Authors tried other multivariate signal techniques to decompose the input EEG signals, such as principal or independent component analysis?

-        How were EEG signals preprocessed? How did the Authors deal with acquisition noise?

-        Settings for EMD decomposition should be indicated.

-        The same applies to the approximate entropy.

-        Have the Authors tried to compute approximate entropy directly from raw EEG data (with no other intermediate steps)?

-        It is not clear which categories classification is referred to. Is it binary (left vs right)? Or multiclass?

-        Figure 8: why IMF energy is not directly used as a discriminant feature?

-        Classification performance should be more accurately assessed, and other parameters than accuracy should be reported: sensitivity, specificity, etc.

-        Table 2: the methods investigated for technical comparisons should be described in the Methods’ section, at least briefly. Before encountering this table, there is no explicit reference to a comparison with the state of the art.

-        Following the previous comment. Classification accuracy from all methods in Table 2 seems comparable for all methods. Are these results statistically significant?

Reviewer 2 Report

This manuscript focus on increasing classification rate in motor imagery based BCI by introducing the feature extraction method based on Discrete Wavelet Transform (DWT), empirical mode decomposition (EMD) and approximate entropy. The method seems interesting, but there are some big defects in the manuscript.

Here are some major things, that must be improved. 
1) The introduction to dataset does not have to be so much, and this data set is well understood by peers. It is recommended to put more space on the introduction of your innovation. 

2) It is highly recommended you use more datasets to prove the efficacy of your approach. Some other datasets, I can suggest are BCI competition datasets like Dataset IIa, BCI Competition IV and Dataset IIIa, BCI Competition III. In addition, Nature has recently published a good data set, suggesting that it can be compared on this new data set. The paper's title is "A large electroencephalographic motor imagery dataset for electroencephalographic brain-computer interfaces".

3) The baseline methods are too old for comparison in table 2. There are some better methods recently published based on IMF, Riemannian manifold and deep learning. The author should compare to some new methods. Here are some, I can suggest are:

(a) Gaur, P.; Pachori, R.B.; Wang, H.; Prasad, G. A multi-class EEG-based BCI classification using multivariate empirical mode decomposition based filtering and Riemannian geometry. Expert Syst. Appl. 2018, 95, 201–211.

(b) P. Wang, A. Jiang, X. Liu, J. Shang, L. Zhang, "LSTM-based EEG classification in motor imagery tasks", IEEE Trans. Neural Syst. Rehabil. Eng., vol. 26, no. 11, pp. 2086-2095, Nov. 2018.

(c) A.Singh, S.Lal and H.W. Guesgen, "Small Sample Motor Imagery Classification Using Regularized Riemannian Features", IEEE Access

(d) Q. She, K. Chen, Y. Ma, T. Nguyen, Y. Zhang, "Sparse representation-based extreme learning machine for motor imagery EEG classification", Comput. Intell. Neurosci., vol. 2018, Oct. 2018.

4) Some references are missing. one example I can suggest like in line 40-41, the author is stating that "However, the essence of these methods is based on Fourier transform, this method cannot achieve good time-frequency resolution at the same time according to the Heisenberg uncertainty principle" 
